# When and How to Lift the Lockdown?
# Global COVID-19 Scenario Analysis and Policy
# Assessment using Compartmental Gaussian Processes

**Zhaozhi Qian**[*]
University of Cambridge
zhaozhi.qian@maths.cam.ac.uk

**Ahmed M. Alaa**[*]
UCLA
ahmedmalaa@ucla.edu

**Mihaela van der Schaar**
University of Cambridge, UCLA, The Alan Turing Institute
mv472@cam.ac.uk

## Abstract

The coronavirus disease 2019 (COVID-19) global pandemic has led many countries to impose unprecedented lockdown measures in order to slow down the outbreak. Questions on whether governments have acted promptly enough, and whether lockdown measures can be lifted soon, have since been central in public discourse. Data-driven models that predict COVID-19 fatalities under different lockdown policy scenarios are essential for addressing these questions and informing governments on future policy directions. To this end, this paper develops a *Bayesian* model for predicting the effects of COVID-19 lockdown policies in a *global* context — we treat each country as a distinct data point, and exploit variations of policies across countries to learn country-specific policy effects. Our model utilizes a *two-layer* Gaussian process (GP) prior — the *lower* layer uses a *compartmental* SEIR (Susceptible, Exposed, Infected, Recovered) model as a prior mean function with "country-and-policy-specific" parameters that capture fatality curves under "counterfactual" policies within each country, whereas the *upper* layer is *shared* across all countries, and learns lower-layer SEIR parameters as a function of a country's features and its policy indicators. Our model combines the solid mechanistic foundations of SEIR models (Bayesian priors) with the flexible data-driven modeling and gradient-based optimization routines of machine learning (Bayesian posteriors) — i.e., the entire model is trained end-to-end via stochastic variational inference. We compare the projections of COVID-19 fatalities in our model with other models listed by the Center for Disease Control (CDC), and provide scenario analyses for various lockdown and reopening strategies highlighting their impact on COVID-19 fatalities.[2]

## 1 Introduction

The ongoing coronavirus disease 2019 (COVID-19) global pandemic poses immense threats not only to public health, but also to the stability of healthcare infrastructures and economies around the world. In an attempt to slow down the outbreak, many countries have imposed unprecedented lockdown and social distancing measures that have eventually proven to be effective in downscaling the volume of COVID-19 fatalities [1, 2, 3] — however, the instigation of such measures gave rise to various *"What if?"* questions that have become central to public discourse, e.g., what the number

---

[*]Equal contribution.
[2]Source code: https://github.com/ZhaozhiQIAN/Compartmental-GP-NeurIPS-2020 or https://bitbucket.org/mvdschaar/mlforhealthlabpub/src/master/.

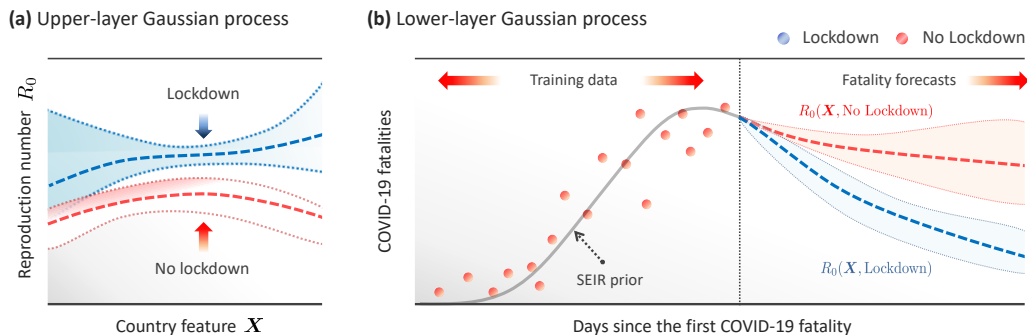

Figure 1: **Pictorial illustration of compartmental Gaussian processes.** (a) The upper-layer GP $f_U$ maps country features and lockdown policies to a predicted $R_0$. Here we depict a simplified binary policy indicator (lockdown or no lockdown). (b) The lower-layer GP $f_L$ maps time to number of COVID-19 fatalities. The mean function is an SEIR model modulated by the upper-layer GP. Projections are obtained using the GP posteriors.

of deaths would have been had the government acted earlier? Would lifting the lockdown cause a second wave of infections? Data-driven models that predict the effects of (lockdown) policies on the trajectory of COVID-19 fatalities are crucial for answering these questions; they are also necessary for informing governments and decision-makers on what policy directions to follow at different stages of the pandemic [4, 5, 6].

In response to calls for data-driven policy-informing models, the academic community has produced various models for forecasting COVID-19 fatalities [7, 8, 9] — these models have been used by public health organizations, such as the WHO and the CDC [28], to advise local authorities on what policy directions to follow. (For instance, the IHME model developed by the University of Washington [7] has been repeatedly cited by the White House in press conferences on COVID-19.[3]) Despite a plethora of epidemiological models for COVID-19 spread, models that can predict the effect of "counterfactual" lockdown policies on COVID-19 fatalities — which is crucial for conducting scenario analyses and policy planning — are still lacking. Moreover, despite the prospects of machine learning (ML) as a key tool for developing such models [10], existing models have been based primarily on epidemiological approaches, with ML techniques being used merely for parameter optimization [11, 12].

In this paper, we develop one of the first ML models for learning lockdown policy effects on COVID-19 fatalities in a *global* context — i.e., we treat each *country* hit by the pandemic as a distinct data point, and exploit the variations in policies followed by different countries to learn country-specific policy effects. We characterize each country with a *feature* vector that comprises economic, social, demographic, environmental and public health indicators; "counterfactual" fatality curves under a hypothetical policy for a given country are thus inferred from "factual" fatality curves of countries "similar" in features wherein this policy was actually implemented. (For example, our model would use data from Sweden to predict what the number of fatalities in Norway would have been under a "herd immunity" policy [13].) We envision that our model would be used by policy-makers to conduct *scenario analyses* by assessing the volume of COVID-19 fatalities under different possible policies — this is especially timely as governments seek policies for gradual lifting of lockdown measures [14].

**How does our model work? How is it different?** A high-level pictorial illustration of our model is provided in Fig. 1. We follow a *Bayesian* approach to jointly model COVID-19 fatalities across many countries through the following hierarchical, two-layer Gaussian process (GP) prior:

$$\textbf{Upper-layer GP:} \quad \boldsymbol{f_U} = R_0(\text{Country features}, \text{Policy indicators}),$$
$$\textbf{Lower-layer GP:} \quad \boldsymbol{f_L} = \text{COVID-19 fatalities over time at a given } R_0.$$

The lower-layer GP $f_L(.)$ models the COVID-19 fatality curve over time within each country — it uses a *compartmental* SEIR (Susceptible, Exposed, Infectious, Recovered) model [15] as its prior mean function, parameterized with the *reproduction number* $R_0$ which characterizes the rate by which the pandemic spreads, i.e., how many people (on average) each new patient infect [16]. The upper-layer GP $f_U(.)$ models the $R_0$ parameter as a function of country features and policy indicators, allowing the model to share data and parameters across different countries that experimented with different policies

(e.g., different lockdown timing). Our model captures uncertainty in both the inferred parameters and the predicted COVID-19 fatalities through the posterior variance across the two layers.

Because of the relative infrequency of pandemics, little related work has been done within the machine learning community to address this problem. In what follows, we provide a brief overview of previous works — an elaborate discussion of the related literature is provided in Appendix A. Previous works prior to the current pandemic have been primarily focused on learning contagion (diffusion) processes on networks, e.g., [17, 18]; unfortunately, these models do not apply to the pandemic as information on social network structures underlying disease spread is hard to obtain. In response to the COVID-19 pandemic, two strands of research work have emerged: (a) methods for devising optimal control (lockdown) policies to contain disease spread [19, 24, 25], and (b) models for forecasting disease spread and expected fatalities [7, 8, 9, 11, 12, 26, 27]. Our paper belongs to category (b) in that the developed model is trained to forecast the future number of fatalities. But unlike existing models in category (b), our model predicts fatalities under different possible lockdown policy choices, hence it can be used for research in category (a) to derive optimal policies.

The most prominent models in category (b) — developed by various academic institutions — have been recognized by the CDC and used to issue national forecasts of COVID-19 fatalities within the United States [28]. Most of these models (e.g., the "UCLA" model [11], the "MIT" model [27] and the IHME model [7]) are SEIR models fit for *individual* countries, with ideas from "machine learning" being used only to optimize the basic SEIR parameters via gradient descent. Our model differs from these models in that it: (1) jointly models fatalities across all of the 170 countries affected by the pandemic, (2) incorporates individual country features to learn how disease dynamics and policy effects vary across countries, (3) enables evaluating future projections under alternative policy scenarios, and (4) combines both mechanistic SEIR models and data-driven machine learning models. Hierarchical modeling has been previously used in the "Imperial" model developed in [26] (which was fit across 11 European countries). This model can be viewed as a special case of ours as it assumes policy effects to be fixed across all countries in its upper layer with no machine learning components to model heterogeneity, and its lower layer uses a serial interval distribution to predict short-term deaths only. Finally, our model extends and unifies the existing works on Gaussian Process models whose mean functions are specified as differential equations or hierarchical models [20, 21, 22, 23]. A detailed table of comparison between all models is provided in Appendix A.

## 2 Problem Setup: Global COVID-19 Scenario Analysis

**Setup and background.** Let $Y_i(t) \in \mathbb{N} \cup \{0\}$ be the number of reported COVID-19 deaths in a given geographical area $i$ on the $t^{th}$ day since the beginning of the outbreak. Throughout this paper, we assume that a geographical area corresponds to a *country*, and consider a set of $N$ countries hit by the pandemic. Each country $i$ is characterized by a feature vector $\boldsymbol{X}_i \in \mathbb{R}^d$ comprising its economic, social, demographic, environmental and public health indicators. Because the number of confirmed daily COVID-19 cases depends greatly on the testing capabilities and strategies within each country [29], we use the reported daily deaths as a more concrete indicator of disease spread.

We model the COVID-19 lockdown *policy* of each country $i$ over days $t$ to $t'$ through the sequence

$$\mathcal{P}_i[t : t'] \triangleq \{p_i(t), \ldots, p_i(t')\}, \tag{1}$$

where $p_i(t) \in \{0, 1\}^K$ is a $K$-dimensional policy indicator variable, reflecting whether country $i$ applies each of $K$ different COVID-19 containment and social distancing measures (e.g., school closure, sheltering in place, travel bans, etc) on day $t$. To build our model, we used data for $N = 170$ countries, each with $d = 35$ features and $K = 9$ policy indicators (See Section 4 and Appendix C for details).

**COVID-19 scenario analysis.** Our key objective is to address the following question: "*How would different lockdown policies affect future COVID-19 fatalities?*" To this end, we build a data-driven model to learn lockdown policy effects on COVID-19 deaths. We envision this model to be trained and applied in a *global* context, where data capturing variations in lockdown policies applied in different countries enables us to learn counterfactual policy effects within each country. Predictions made by our model can be used to conduct scenario analyses that inform government policy.

To build our model, we consider a data set $\mathcal{D}_{N,t}$ for $N$ countries covering a period of $t$ days, i.e.,

$$\mathcal{D}_{N,t} \triangleq \{\boldsymbol{X}_i, \mathcal{Y}_i[1 : t], \mathcal{P}_i[1 : t]\}_{i=1}^{N}, \tag{2}$$

where $\mathcal{Y}_i[t:t'] \triangleq \{Y_i(t), \ldots, Y_i(t')\}$. For each country $i$, we forecast the trajectory of the number of COVID-19 deaths within a future time horizon of $T$ days under a given lockdown policy, i.e.,

$$\widehat{\mathcal{Y}}_i[t:t+T] = \mathbb{E}\Big[\mathcal{Y}_i[t:t+T] \mid \underbrace{p_i(t), p_i(t+1), \ldots, p_i(t+T)}_{\text{Future policy } \mathcal{P}_i[t:t+T]}, \mathcal{D}_{N,t}\Big]. \tag{3}$$

In addition to point predictions, we also estimate a sequence of uncertainty intervals $\widehat{\mathcal{C}}_i[t:t+T]$ that cover the true trajectory of future COVID-19 fatalities, i.e., $\mathcal{Y}_i[t:t+T]$, with high probability.

## 3 Compartmental Gaussian Processes

In this Section, we develop a *Bayesian* model for the effects of lockdown policies on future COVID-19 fatalities. We provide our model's specification in Section 3.1, and then develop the corresponding model training and parameter learning algorithms in Section 3.2.

### 3.1 Model Specification

**Two-layer GP prior.** We assume a (hierarchical) two-layer Gaussian process (GP) [30, 31] prior on the COVID-19 fatality curves $\{Y_i(t)\}_i$ for the $N$ countries under consideration as follows:

$$\begin{aligned}
\textbf{Upper-layer GP:} \quad & f_U \sim \mathcal{GP}(m_\alpha(\boldsymbol{X}, p), K_\alpha((\boldsymbol{X}, p), (\boldsymbol{X}', p'))), \\
\textbf{Lower-layer GP:} \quad & f_{L,i} \sim \mathcal{GP}(D_{\theta_i}(t), K_{\theta_i}(t, t')), \ \theta_i = v(f_U(\boldsymbol{X}_i, p_i)),
\end{aligned} \tag{4}$$

for $i \in \{1, \ldots, N\}$, where $v(.)$ is a transformation function (described later this Section), $\alpha$ and $\theta_i$ are the hyper-parameter sets for the upper- and lower-layer (associated with country $i$) GPs, respectively. The model in (4) assumes that the fatality curve $Y_i(t)$ for each country $i$ is a noisy draw from a GP prior with a mean function $D_{\theta_i}$ and kernel function $K_{\theta_i}$, which implies that $Y_i(t)$ follows a Gaussian distribution. The prior on lower-layer parameters $\theta_i$ is specified via the upper-layer GP — the function $f_U$ maps country $i$'s features $\boldsymbol{X}_i$ and its adopted policy $p_i$ to a parameter $\theta_i$[4]. The mean function $m_\alpha$ for the upper-layer GP is assumed to be a constant, and an RBF kernel is selected for both layers. Because the upper-layer GP shares its parameters $\alpha$ across all countries, countries with "similar" features and policies will share similar parameters, and hence similar fatality profiles. The graphical model for (4) is provided in Fig. 2 (a).

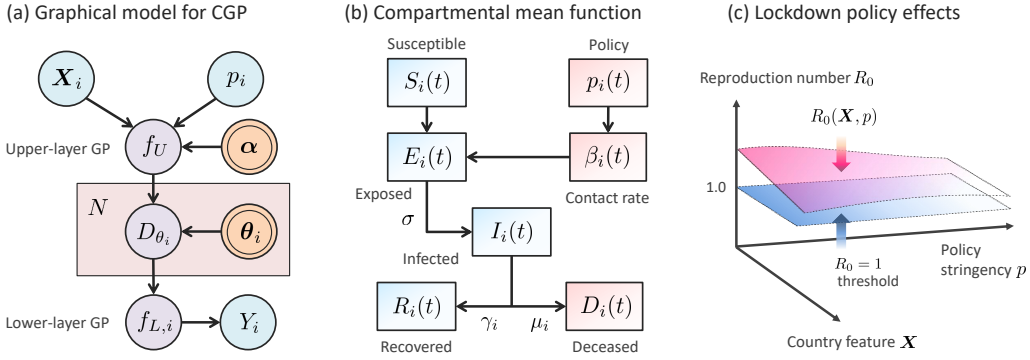

Figure 2: **Illustration for the CGP model.** (a) Graphical model for CGP. (b) Block diagram for the SEIR model underlying the CGP compartmental prior. (c) Policy effects as function of country features and policy indicators.

**Compartmental priors.** With little data available in the early stages of the pandemic, our prior knowledge on the expected COVID-19 fatality curves are limited to mathematical models for the spread of infectious diseases. Thus, we model the prior mean function $D_{\theta_i}(t)$ in (4) via a compartmental *SEIR* (Susceptible, Exposed, Infectious, Recovered) model [32], with parameters $(\beta_i(t), \sigma, \gamma_i, \mu_i)$, i.e.,

$$D_{\theta_i}(t) = \text{SEIR}(\beta_i(t), \sigma, \gamma_i, \mu_i), \tag{5}$$

where $\beta_i$ is the *contact rate* (average number of contacts per person per unit time), $\sigma$ is the virus's *incubation rate*, $\gamma_i$ is the patient *recovery rate*, and $\mu_i$ is the *mortality rate*. With the exception of $\sigma$, all of the other SEIR parameters are country-specific since they depend on population demographics (e.g., contact rates depend on social mobility and mortality rates depend on age distribution [33]). The parameter set $\theta_i$ for the lower-layer GP comprises the SEIR parameters, in addition to the length-scale $\ell_L$ and the variance $\eta$ of the kernel function in (4), i.e., the lower-layer parameter set is $\theta_i = (\ell_L, \eta, \beta_i(t), \sigma, \gamma_i, \mu_i)$.

In an SEIR model, individuals within the population of country $i$ are assigned to 5 compartments: Susceptible $S_i(t)$, Exposed $E_i(t)$, Infectious $I_i(t)$, Recovered $R_i(t)$ or Deceased $D_i(t)$ — each individual progresses through these compartments as dictated by the following differential equations [35, 36]:

$$\frac{dS_i}{dt} = \mu_i(n_i - S_i) - \frac{\beta_i S_i I_i}{n_i}, \quad \frac{dE_i}{dt} = \frac{\beta_i S_i I_i}{n_i} - (\mu_i + \sigma)\, E_i, \quad \frac{dI_i}{dt} = \sigma\, E_i - (\gamma_i + \mu_i)\, I_i, \quad (6)$$

with $dR_i/dt = \gamma_i\, I_i - \mu_i\, R_i$, and $dD_i/dt = \mu_i\, I_i$, where $n_i$ is the population size of country $i$. In (6), we assume that each country's contact rate $\beta_i(t)$ is *time-dependent* in order to account for the impact of (lockdown) policy changes over time. The death compartment $D_i$ obtained by solving (6) is set as the prior mean function $D_{\theta_i}(t)$ for the lower-layer GP in (4). Our choice of the SEIR model as the prior fatality curve is motivated by the dynamics of the SARS-CoV-2 virus that causes COVID-19; since the SEIR model captures incubation rates [36], it more accurately represents SARS-CoV-2 dynamics — compared to other SIR variants — which are known to exhibit significant incubation periods [37]. Note that while we select an SEIR model as our prior, other compartmental models can be used as well. A block diagram describing the compartmental prior mean function is given in Fig. 2 (b).

The Bayesian nature of our model enables combining the rigorous mechanistic foundation of compartmental models with the data-driven (nonparameteric) nature of GPs. That is, in the early stages of the pandemic, the early fatality forecasts would be dominated by the prior mean function $D_{\theta_i}(t)$ derived from the SEIR prior — as more data on COVID-19 fatalities are collected over time, the GP posterior will refine the SEIR forecasts based on observed patterns in the data. Because of its hybrid nature, we call our model a *compartmental Gaussian process* (CGP).

**Modeling policy effects.** Governments decide on (lockdown) policies over time in order to control the basic reproduction number $(R_0)$ of COVID-19, i.e., the rate by which the pandemic spreads [16]. The reproduction number within country $i$ as represented by the SEIR prior in (6) is given by [35]:

$$R_{0,i}(t) = \frac{\sigma}{\mu_i + \sigma} \cdot \frac{\beta_i(t)}{\mu_i + \gamma_i}, \qquad (7)$$

where $R_{0,i}(t)$ is the reproduction number in country $i$ at time $t$. An $R_{0,i}(t)$ of a value less than 1 means that country $i$'s fatality curve $Y_i(t)$ is "flattening" — the policy $p_i(t)$ aims at driving $R_{0,i}(t)$ below 1 by imposing social distancing measures that would minimize the contact rate $\beta_i(t)$ in (7).

Because different countries with comparable features apply different policies, the upper-layer GP function $f_U(\boldsymbol{X}, p)$ in (4) will learn counterfactual fatality curves for each country under alternative policies that have been tried in other countries (See Fig 2(c)). For instance, we can learn the fatality curves for Scandinavian countries (such as Norway and Denmark) under a hypothetically less restrictive lockdown policy using data from Sweden, which adopted less stringent policy measures [13].

The lower-layer parameters are linked to the upper layer as follows: $f_U$ is specified as a multi-output GP; each output corresponding to a distinct parameter in $\theta_i = (\ell_L, \beta_i(t), \sigma, \gamma_i, \mu_i)$. The transformation function $v(.)$ in (4) is an identity map for all outputs corresponding to parameters $(\ell_L, \sigma, \gamma_i, \mu_i)$, except for the contact rate parameter $\beta_i(t)$ where it performs the following mapping:

$$\beta_i(t) = v(f_U(\boldsymbol{X}_i, p_i(t))) = 2\,\bar{\beta}\,\mathrm{Sigmoid}(f_U(\boldsymbol{X}_i, p_i(t))), \qquad (8)$$

where $\bar{\beta}$ is a reference value for the contact rate obtained using early data from Wuhan province [38].

**Scenario analysis via posterior inference.** What is the expected number of COVID-19 fatalities in country $i$ given a future lockdown policy scenario $\mathcal{P}_i[t : t+T]$? To answer this question, we compute the posterior distribution of $\mathcal{Y}_i[t : t+T]$ given the data $\mathcal{D}_{N,t}$ and the policy scenario $\mathcal{P}_i[t : t+T]$, i.e.,

$$\mathbb{P}\big(\mathcal{Y}_i[t : t+T]\,\big|\,\mathcal{P}_i[t : t+T]\big) = \int \underbrace{\mathbb{P}_\theta\big(\mathcal{Y}_i[t : t+T]\,\big|\,\mathcal{Y}_i[1 : t]\big)}_{\textbf{Lower layer}} \cdot \underbrace{d\mathbb{P}_\alpha\big(\theta\,\big|\,\mathcal{D}_{N,t}, \mathcal{P}_i[t : t+T]\big)}_{\textbf{Upper layer}},$$

where conditioning on $\mathcal{D}_{N,t}$ in the left hand side was omitted for brevity. Here, the lower- and upper-layer posteriors are computed analytically using the closed-form expression for GP posteriors [30], and the integral is evaluated via a Monte Carlo approximation. Point estimates and uncertainty intervals are obtained by evaluating the mean and variance of the resulting distribution.

## 3.2 Learning via Stochastic Variational Inference

Accurate posterior inferences of $\mathcal{Y}_i[t : t+T]$ require training the CGP model by optimizing the parameter $\alpha$ of the upper-layer GP using the observed data $\mathcal{D}_{N,t}$ by maximizing the model's log-likelihood:

$$\mathcal{L}(\mathcal{D}_{N,t} \,|\, \alpha) \triangleq \log \int \prod_{i=1}^{N} \mathbb{P}\big( \mathcal{Y}_i[1 : t] \,\big|\, \theta_i \big) \cdot \mathbb{P}\big( \theta_i \,\big|\, \boldsymbol{X}_i, \mathcal{P}_i[1 : t], \alpha \big) \, d\theta_i, \qquad (9)$$

with $\alpha^* = \arg\max_\alpha \mathcal{L}(\mathcal{D}_{N,t} \,|\, \alpha)$. Because the integral in (9) is intractable, we resort to a variational inference approach for optimizing the model's likelihood [39, 40, 41, 42]. That is, to train our model, we minimize the Evidence lower bound (ELBO) on (9) given by:

$$\log \mathbb{P}\big( \mathcal{Y}_i[1 : t] \,|\, \alpha \big) \geq \mathrm{ELBO}_i(\alpha, \phi) = \mathbb{E}_{\mathbb{Q}}\big[ \log \mathbb{P}\big( \mathcal{Y}_i[1 : t], \theta_i \,|\, \alpha \big) - \log \mathbb{Q}\big( \theta_i \,|\, \mathcal{Y}_i[1 : t], \phi \big) \big],$$

where $\mathbb{Q}(.)$ is the variational distribution with parameters $\phi$, and conditioning on $\boldsymbol{X}_i$ and $\mathcal{P}_i[1 : t]$ is suppressed for notational brevity. We choose a normal distribution for $\mathbb{Q}(.)$ — this renders analytic evaluation of the ELBO objective and its gradients possible. We use stochastic gradient descent via ADAM algorithm to optimize the ELBO objective [43], and update the lower-layer parameters in each gradient iteration by solving the SEIR differential equations in (6) using Euler's method [44]. A pseudo-code for our training algorithm is provided in Appendix B.

# 4 Experiments

In this Section, we use our CGP model to forecast COVID-19 fatalities in various countries around the world, taking into account the lockdown policies within these countries. We compare the projections of our model with other models listed by the Center for Disease Control (CDC) (Section 4.2), and show how our model can be used to analyze counterfactual policy scenarios (Section 4.3).

## 4.1 Experimental Setup

**Data description.** We collated the data set $\mathcal{D}_{N,t} = \{\boldsymbol{X}_i, \mathcal{Y}_i, \mathcal{P}_i\}_{i=1}^{N}$ for $N = 170$ countries affected by the COVID-19 pandemic using three data sources: (1) published World Bank reports[5] were used to extract a set of $d = 35$ features $\boldsymbol{X}_i$ for each country, (2) the COVID-19 CSSE data repository at Johns Hopkins University [45] was used to extract each country's fatality time-series $\mathcal{Y}_i$, and (3) the Oxford COVID-19 Government Response Tracker (OxCGRT) — curated by the Blavatnik School of Government at Oxford University [46] — was used to extract $K = 9$ policy indicators $\mathcal{P}_i$ for each country over time. Our data set covered the period spanning from January 22, 2020 to May 8, 2020.

Each country's features included a comprehensive set of economic, demographic, environmental, social and health indicators (e.g., population density, prevalence of obesity, air transport frequency, median age, prevalence of hand-washing facilities, health expenditure, etc). Policy indicators included: information on school and workplace closure, public events' cancellation, travel restrictions, public transport closure, etc. A complete list of variables included in our model is provided in Appendix C.

**Implementation.** We implemented our CGP model using `Pyro` [47], a universal probabilistic programming language in Python supported by a PyTorch backend. The variational inference algorithm in Section 3.2 was implemented using ADAM with 1000 iterations and a learning rate of 0.01. Further details on hyper-parameter tuning is provided in Appendix C. Projections from all baselines involved in our comparisons were obtained from the official CDC website [28].

**Baselines.** We considered the most prominent baseline models listed by the CDC [28]: the "UCLA" model [11], the "MIT-DELPHI" model [27], the Los Alamos National Laboratory "LANL" model [50], the "Imperial" model [26], the IHME model [7], the "YYG" model [12] which was found to be the best performing CDC-listed model in recent weeks [49], in addition to

| Model | March 28 forecasts (Before the peak) | | April 11 forecasts (During the peak) | | April 25 forecasts (After the peak) | |
|---|---|---|---|---|---|---|
| | 7 days | 14 days | 7 days | 14 days | 7 days | 14 days |
| YYG | — | — | -6,470 | -10,528 | -662 | -1,458 |
| Imperial | -881 | — | **-1,757** | — | **14** | — |
| LANL | — | — | -6,010 | **-3,161** | -2,018 | -2,989 |
| MIT-DELPHI | — | — | — | — | -2,054 | **549** |
| Gompertz curve | — | — | 2,174 | 4,689 | -2,728 | -7,062 |
| Vanilla SEIR | 2,723 | 4,822 | -11,328 | -24,189 | -9,696 | -21,314 |
| IHME | -1,999 | -2,289 | -6,134 | -10,129 | -3,623 | -9,999 |
| CDC-ensemble | — | — | -2,739 | -8,188 | -4,244 | -5,091 |
| CGP (US data only) | **-642** | -4,380 | -3,182 | -8,260 | -560 | -881 |
| CGP | -867 | **-1,396** | -1,906 | -4,518 | -439 | 611 |

Table 1: Accuracy of predicted cumulative deaths at different stages of the pandemic in the United States.

| Country | Mean Absolute Error on Daily Deaths (CRPS: continuous ranked probability score) | | | | | | |
|---|---|---|---|---|---|---|---|
| | 14-day Forecasts | | | | 30-day Forecasts | | |
| | CGP | Imperial | IHME | YYG | CGP | IHME | YYG |
| USA | 139 (.076) | 149 (.282) | 753 (.164) | **50** (**.073**) | 481 (.196) | 957 (.260) | **365** (**.164**) |
| GBR | **58** (.089) | 164 (.248) | 288 (**.088**) | 178 (.224) | 231 (.291) | 259 (**.156**) | **140** (.176) |
| ITA | 78 (**.090**) | **63** (.226) | 202 (.298) | 87 (.192) | **55** (**.119**) | 179 (.324) | 90 (.184) |
| DEU | **30** (**.100**) | 51 (.247) | 54 (.151) | 70 (.249) | **45** (**.197**) | 46 (.230) | 91 (.273) |
| ESP | 125 (**.121**) | 88 (.236) | 133 (.197) | **82** (.183) | 83 (**.168**) | 140 (.273) | **81** (.170) |
| FRA | **26** (**.075**) | 85 (.239) | 148 (.216) | 124 (.161) | **104** (.190) | 150 (.282) | 153 (**.170**) |
| NLD | **11** (.131) | 29 (.298) | 83 (**.112**) | 34 (.220) | **32** (.277) | — | 45 (**.241**) |
| SWE | **11** (.098) | 34 (.271) | 35 (**.082**) | 32 (.218) | **34** (**.210**) | 118 (**.210**) | 38 (.228) |
| PRT | **1** (**.092**) | 2 (.176) | 7 (.186) | 10 (.260) | **3** (**.174**) | 10 (.275) | 12 (.263) |

Table 2: Accuracy of predicted daily deaths and uncertainty estimates over different time horizons. (The Imperial model does not provide 30-day forecasts.)

the CDC-ensemble, which issues a national-level forecast for the United States by combining the predictions of 16 individual models [28]. The UCLA, MIT-DELPHI, LANL, YYG and IHME models are all variants of the SEIR model — the IHME is arguably the most influential of these models, having been often cited during White House press briefings on COVID-19 modeling efforts [51]. In addition to these models, we consider a vanilla SEIR model and a standard Gompertz curve fitting approach as baselines [48].

### 4.2 Forecasting Accuracy

**Forecasting Accuracy for the United States.** We start off by comparing the accuracy of fatality projections by our model with the baselines for the United States. In Table 1, we evaluate the accuracy of the 7-day and 14-day projections issued at three stages of the pandemic: before the peak of infections (March 28), in the midst of the peak (April 11) [45], and in the "plateauing" stage (April 25). Accuracy was evaluated by computing the difference between true and predicted cumulative deaths throughout the forecasting horizon, i.e., $\sum_{k=1}^{T}(Y_i(t+k) - \widehat{Y}_i(t+k))$ for $T = 7$ and $T = 14$. We evaluate the models based on *cumulative* death *within* the horizon because at the early stage of the pandemic, none but the IHME model issues *daily* predictions for two weeks. For each forecast, only data preceding the forecast date was used for training our model. The results are summarized in Table 1, with the best performing model in each forecast highlighted via an underlined bold font.

As we can see in Table 1, CGP outperforms the CDC-ensemble and the prominent IHME models at almost all forecasts; most notably the CGP predictive accuracy is an order of magnitude better in the plateau stage of the pandemic (around April 25). Our model also performs consistently well across all

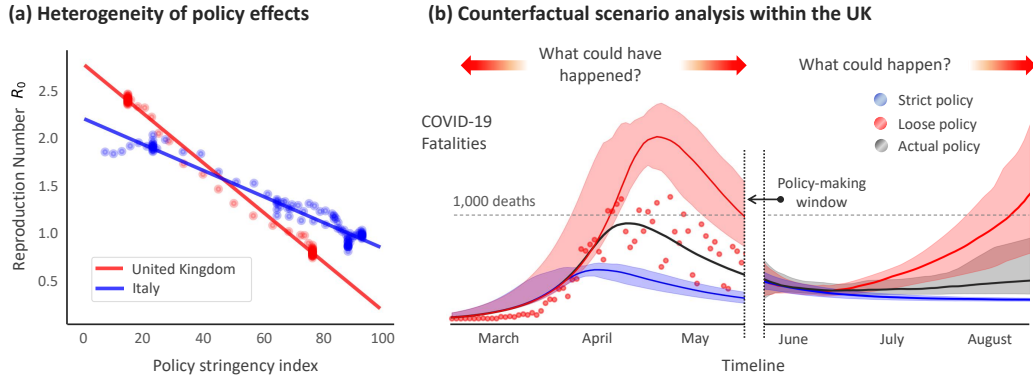

**(a) Heterogeneity of policy effects**

**(b) Counterfactual scenario analysis within the UK**

Figure 3: **Policy effects learned by the CGP model.** (a) Regression slopes for the impact of lockdown policy stringency on $R_0$. (b) Counterfactual COVID-19 fatality curves for the UK under different lockdown policies.

forecasts — its performance is comparable to the best model for each forecast. Note that, as mentioned earlier in Section 1, the Imperial model provides a competitive performance on short-term predictions because it is tailored to inferences based on short-term transmission dynamics, but it falls short when it comes to long-term forecasts, which can be crucial for anticipating the timing of the peak. Last but not least, CGP uses the policy indicators $\mathcal{P}_i$ to inform its forecast. The ability to incorporate policy decisions into the forecast is a unique feature of CGP and it likely contributes to the performance gain. To the best of our knowledge, most benchmarks only consider the historical fatality time series $\mathcal{Y}_i$ (with the exception of YYG, which also uses the underground ridership data [12]).

The benefits of joint (global) modeling across multiple countries are demonstrated through an ablated version of our model that uses United States (US) data only for training. The global version of our model consistently achieves significant performance gains compared to the version that uses US data only; similar patterns were found in other countries (See Appendix C). This shows the value of using machine learning to capture country-specific disease spread parameters based on *country features $X_i$*. In comparison, neither the ablated version nor the benchmarks uses such country level features.

**Accuracy of global projections.** In Table 2, we compare the forecasting accuracy of our model with the CDC-listed baselines that offer projections for countries other than the US (IHME, Imperial and YYG). We evaluated the performance of all baselines in 9 countries that were significantly affected by the pandemic. Additional results on other countries in Europe, Asia, Africa, and the Americas are available in Appendix C3. (Note that ours is the only model that covers all countries and spans all continents.) Since these countries were at different stages of the pandemic at any given time, we evaluated the 7-day and 14-day forecasts on April 25, 2020, when all countries have had a significant number of infections. Accuracy was evaluated by computing the mean *absolute* error on predicted *daily* deaths: $\mathcal{E} = \frac{1}{T}\sum_{k=1}^{T} |Y_i(t+k) - \widehat{Y}_i(t+k)|$ for $T = 14$ and $T = 30$. We evaluated the quality of uncertainty measures in terms of the average continuous ranked probability score (CRPS) [34]. As we can see in Table 2, our model outperforms the baselines in almost all countries on both forecasting horizons. Central to the accuracy of our model is its hierarchical GP structure which shares data across countries based on their "feature similarity", enabling accurate predictions even when little data is available for individual countries.

Our model does not only learn the fatality curves within each country, but also the country-specific effects of lockdown. Since the same policy would naturally yield different effects in different countries, this is a major source of performance gain for our model compared to others (e.g., Imperial model) which assume fixed policy effects. To demonstrate the heterogeneity of policy effects, Fig. 3 (a) shows the inferred $R_0$ in the United Kingdom (UK) against the stringency of the policy applied at different points of time. We quantify the policy stringency through the *stringency index* defined in [46], which collapses all policy indicators into a single number between 0 and 100 — a higher index corresponds to a stricter policy. The regression slope in Fig. 3(a) reflects the policy effects, i.e., how much a unit increase in policy stringency reduces $R_0$. As we can see, the lockdown effects learned by our model differ among countries based on their features; here we compare policy effects for the UK and Italy as an exemplar. We specify the country features most relevant to policy effects in Appendix C.

### 4.3 Counterfactual Analysis

As we have seen in Section 4.2, our model performs favorably compared to existing models in terms of predicting COVID-19 fatalities, but how can it be used to inform decision-making? Here, we use our model to analyze the lockdown policy of the UK in the early stages of the pandemic, which sparked controversies within the British society [52, 53, 54], and provide projections for future COVID-19 fatalities under the gradual re-opening plan announced by the UK government [55].

In Fig. 3 (b) we display the predictions and counterfactual inferences of our model from the policy-maker's perspective in the period spanning from May 8 until the end of May, when a lockdown lifting policy was being planned for. At this point of time, the policy-maker can be presented with counter-factual fatality curves of what would have happened before May 8 had lockdown been implemented **1 week earlier (blue curve)** or **1 week later (red curve)**. Our model predicts that 13,827 lives would have been saved with an earlier lockdown, and 22,405 more deaths would have occured under a later one. Looking into the future, our model predicts that under the current UK government re-opening plan **(black curve)**, daily deaths would stabilize around 200. Maintaining the current lockdown **(blue curve)** would lead daily deaths to fall under 100 in August, which would save 6,215 more lives compared to the current re-opening plan. On the other hand, a complete re-opening in June **(red curve)** would result in a second peak in August-September although there is substantial uncertainty about the volume of the second peak. Similar analyses for other countries are given in Appendix C.

We envision our model to be used by governments to make measured decisions that might impact the lives of millions of people all over the world. We hope that our model exemplifies the importance of machine learning-based decision-making for public health in the post-coronavirus world.

## Broader Impact

This paper addresses a timely decision-making problem that faces governments and authorities around the world during these exceptional times. Decisions informed by our model may affect the daily lives of millions of people around the world during the upcoming months. We believe that now is the time for research on machine learning for clinical and public health applications to contribute to the efforts humanity exerts to handle the current crisis — we hope that our model plays a role in informing the public and governments on the consequences of policies and social behavior on public health. We are currently in the phase of communicating the projections of our model with official public health services in multiple countries, including developing countries.

## Acknowledgments and Disclosure of Funding

This work was supported by the Office of Naval Research (ONR) and the NSF (Grant number: 1722516).

## Footnotes

[3]edition.cnn.com/2020/05/20/health/model-on-masks-coronavirus

[4]Additional sources of information, such as the level of domestic and international travel, were not consistently available across the globe at the time of writing and therefore they were not included in our model.

[5] https://data.worldbank.org/

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
