[Supplementary Material]

# Appendix for "When and How to Lift the Lockdown? Global COVID-19 Scenario Analysis and Policy Assessment using Compartmental Gaussian Processes"

**Zhaozhi Qian**[*]
University of Cambridge
zhaozhi.qian@maths.cam.ac.uk

**Ahmed M. Alaa**[*]
UCLA
ahmedmalaa@ucla.edu

**Mihaela van der Schaar**
University of Cambridge, UCLA, The Alan Turing Institute
mv472@cam.ac.uk

## Appendix A: Related Literature

In this Section, we provide a detailed comparison between different existing approaches for modeling fatality curves. A tabulated comparison between our model and existing ones is laid out in Table A1.

| Approach | Training Approach | Modeling Scope | Policy Effects | Model Uncertainty |
|---|---|---|---|---|
| Compartmental models | Exhaustive parameter search | Individual countries | Not incorporated | None |
| Machine learning-based compartmental models | Gradient descent optimization | Individual countries | Not incorporated | Ad-hoc bootstrap estimates |
| Bayesian mechanistic hierarchical model | Posterior inference via MCMC methods | Individual countries | Incorporated | Bayesian credible intervals |
| Curve fitting | Exhaustive parameter search | Individual countries | Not incorporated | None |
| Dynamic control modeling | Model parameters based on expert knowledge | Individual countries | Incorporated | None |
| Recurrent neural networks for fatality curve modeling [49] | Gradient descent optimization | Individual countries | Not incorporated | None |
| **Compartmental Gaussian processes** | **Stochastic gradient-based variational inference** | **Global** | **Incorporated** | **Bayesian credible intervals** |

Table A1: Comparison between existing approaches to model fatality curves.

Because of the relative infrequency of pandemics, little related work has been done within the machine learning community to address this problem. In what follows, we provide a brief overview of previous works. Previous works prior to the current pandemic have been primarily focused on learning contagion (diffusion) processes on networks, e.g., [15, 16]; unfortunately, these models do

---

[*]Equal contribution.

not apply to the pandemic as information on social network structures underlying disease spread is hard to obtain. In response to the COVID-19 pandemic, two strands of research work have emerged: (a) methods for devising optimal control (lockdown) policies to contain disease spread [17, 18, 19], and (b) models for forecasting disease spread and expected fatalities [5, 6, 7, 9, 10, 20, 21]. Our paper belongs to category (b) in that the developed model is trained to forecast the future number of fatalities. But unlike existing models in category (b), our model predicts fatalities under different lockdown policy choices, hence it can be used for research in category (a) to derive optimal policies.

The most prominent models in category (b) — developed by various academic institutions — have been recognized by the CDC and used to issue national forecasts of COVID-19 fatalities within the United States [22]. Most of these models (e.g., the "UCLA" model [9], the "MIT" model [21] and the IHME model [5]) are SEIR models fit for *individual* countries, with ideas from "machine learning" being used only to optimize the basic SEIR parameters via gradient descent. Our model differs from these models in that it: (1) jointly models fatalities across all of the 170 countries affected by the pandemic, (2) incorporates individual country features to learn how disease dynamics and policy effects vary across countries, (3) enables evaluating future projections under alternative policy scenarios, and (4) combines both mechanistic SEIR models and data-driven machine learning models. Hierarchical modeling has been previously used in the "Imperial" model developed in [20] (which was fit across 11 European countries). This model can be viewed as a special case of ours as it assumes policy effects to be fixed for all countries in its upper layer with no machine learning components to model heterogeneity, and its lower layer uses an interval distribution to predict short-term deaths only.

## Appendix B: Learning via Stochastic Variational Inference

Accurate posterior inferences of $\mathcal{Y}_i[t : t+T]$ require training the CGP model by optimizing the parameter $\alpha$ of the upper-layer GP using the observed data $\mathcal{D}_{N,t}$ by maximizing the model's log-likelihood:

$$\mathcal{L}(\mathcal{D}_{N,t} \,|\, \alpha) \triangleq \log \int \prod_{i=1}^{N} \mathbb{P}\big(\mathcal{Y}_i[1 : t] \,\big|\, \theta_i\big) \cdot \mathbb{P}\big(\theta_i \,\big|\, \boldsymbol{X}_i, \mathcal{P}_i[1 : t], \alpha\big) \, d\theta_i, \tag{1}$$

with $\alpha^* = \arg\max_\alpha \mathcal{L}(\mathcal{D}_{N,t} \,|\, \alpha)$. Because the integral in (1) is intractable, we resort to a variational inference approach for optimizing the model's likelihood [32, 33, 34, 35]. That is, to train our model, we minimize the Evidence lower bound (ELBO) on (1) given by:

$$\log \mathbb{P}\big(\mathcal{Y}_i[1 : t] \,\big|\, \alpha\big) \geq \text{ELBO}_i(\alpha, \phi) = \mathbb{E}_{\mathbb{Q}}\left[\log \mathbb{P}\big(\mathcal{Y}_i[1 : t], \theta_i \,\big|\, \alpha\big) - \log \mathbb{Q}\big(\theta_i \,\big|\, \mathcal{Y}_i[1 : t], \phi\big)\right],$$

where $\mathbb{Q}(.)$ is the variational distribution with parameters $\phi$, and conditioning on $\boldsymbol{X}_i$ and $\mathcal{P}_i[1 : t]$ is suppressed for notational brevity. We choose a normal distribution for $\mathbb{Q}(.)$ — this renders analytic evaluation of the ELBO objective and its gradients possible. We use stochastic gradient descent via ADAM algorithm to optimize the ELBO objective [36], and update the lower-layer parameters in each gradient iteration by solving the SEIR differential equations using Euler's method [37].

The pseudo-code for the learning algorithm is provided below:

1. Sample $\theta_i^{(j)} \sim \mathbb{Q}\big(\theta_i \,\big|\, \mathcal{Y}_i[1 : t], \phi\big)$, $i = 1, \ldots, N, j = 1, \ldots, m$.
2. Estimate $\mathcal{L}(\mathcal{D}_{N,t} \,|\, \alpha) = \log \sum_{j=1}^{m} \prod_{i=1}^{N} \mathbb{P}\big(\mathcal{Y}_i[1 : t] \,\big|\, \theta_i^{(j)}\big) \cdot \mathbb{P}\big(\theta_i^{(j)} \,\big|\, \boldsymbol{X}_i, \mathcal{P}_i[1 : t], \alpha\big) \, d\theta_i^{(j)}$.
3. Estimate the gradients $\nabla_\theta \mathcal{L}$ and $\nabla_\phi \mathcal{L}$.
4. Solve the SEIR differential equations using Euler's method.
5. Update $\theta$ and $\phi$.

## Appendix C: Experiments

### Model Variables

We collated the data set $\mathcal{D}_{N,t} = \{\boldsymbol{X}_i, \mathcal{Y}_i, \mathcal{P}_i\}_{i=1}^{N}$ for $N = 170$ countries affected by the COVID-19 pandemic using three data sources: (1) published World Bank reports were used to extract a set of $d = 35$ features $\boldsymbol{X}_i$ for each country, (2) the COVID-19 CSSE data repository at Johns Hopkins University [38] was used to extract each country's fatality time-series $\mathcal{Y}_i$, and (3) the Oxford COVID-19 Government Response Tracker (OxCGRT) — curated by the Blavatnik School of Government at

Oxford University [39] — was used to extract $K = 9$ policy indicators $\mathcal{P}_i$ for each country over time. Our data set covered the period spanning from January 22, 2020 to May 8, 2020.

Each country's features included a comprehensive set of economic, demographic, environmental, social and health indicators (e.g., population density, prevalence of obesity, air transport frequency, median age, prevalence of hand-washing facilities, health expenditure, etc). Policy indicators included: information on school and workplace closure, public events' cancellation, travel restrictions, public transport closure, etc. All variables inlcuded in our model are listed in Tables C1 and C2.

| $I^0$: School closure | $I^1$: Stay-at-home requirements | $I^2$: Restrictions on gathering size |
|---|---|---|
| $I^3$: Workplace closure | $I^4$: Restrictions on domestic or internal movement | $I^5$: Public transport closures |
| $I^6$: Cancellation of public events | $I^7$: Restrictions on international travel | $I^8$: Public information campaign |

Table C1: Individual policy indicators used in our model.

**Economic Indicators**

GDP per capita, GNI per capita, Income share held by lowest 20%

**Social and Demographic Indicators**

Population, Life expectancy, Birth rate, Death rate, Infant mortality rate, Land Area,
% People with basic hand-washing facilities including soap and water, Smoking prevalence,
Prevalence of undernourishment, Prevalence of overweight, Urban population,
Population density, Population ages 65 and above, Access to electricity (% of population),
UHC service coverage index, Total alcohol consumption per capita,
Air transport (passengers carried)

**Environmental Indicators**

Forest Area, PM2.5 air pollution (mean annual exposure in micrograms per cubic meter)

**Public Health Indicators**

Immunization for measles, % deaths by communicable diseases, Current health expenditure,
Current health expenditure per capita, Diabetes prevalence, Immunization for DPT,
Immunization for HepB3, Incidence of HIV, Incidence of malaria, Incidence of tuberculosis,
% deaths by CVD/cancer/diabetes/CRD , % deaths due to household and ambient air pollution,
% deaths due to unsafe water/unsafe sanitation/lack of hygiene, Physicians (per 1,000 people)

Table C2: Economic, social, demographic, environmental and health indicators for each country considered in our analysis. Data on these indicators was obtained from the World Bank (https://data.worldbank.org/).

**Model Implementation Details**

We implemented our CGP model using `Pyro` [40], a universal probabilistic programming language in Python supported by a PyTorch backend. The variational inference algorithm was implemented using ADAM with 1000 iterations and a learning rate of 0.01. The Gaussian process hyper-parameters were tuned by keeping the last 14 days worth of fatality data as a validation set, and grid search was used to tune the kernel hyper-parameters (length-scale). Projections from all baselines involved in our comparisons were obtained from the official CDC website [22].

**Results**

In this Section, we provide further experimental results in addition to those presented in Section 4. In what follows, we list the results provided in this appendix.

- **Table C3**: Performance of our model with and without cross-country joint modeling.
- **Table C4**: Correlation between country features and effect of lockdown on $R_0$.
- **Table C5**: Correlation between country features and $R_0$ before lockdown.
- **Table C8**: The difference in total deaths by April 25 as reported at April 25 and May 8.
- **Figure C6**: Policy stringency index (defined by te Oxford policy tracker [3]) and $R_0$ over time within different countries.
- **Figure C7**: Counterfactual scenario analysis for France. The blue curve corresponds to the current lockdown measures continuing, whereas the red curve corresponds to the current re-opening plan.

| Country | 4/25 - 5/01 (One week) | | | | | 4/25 - 5/07 (Two weeks) | | | | |
|---|---|---|---|---|---|---|---|---|---|---|
| | IHME | YYG | Imperial | CGP (local) | CGP (global) | IHME | YYG | Imperial | CGP (local) | CGP (global) |
| Austria | -26 | 15 | 1 | | -37 | - | 55 | 20 | | -39 |
| Brazil | - | -283 | - | 454 | -105 | - | -768 | - | 3011 | -298 |
| Denmark | 37 | 10 | 14 | | -5 | - | 2 | 4 | | -10 |
| **Egypt** | - | - | - | -6 | -47 | - | - | - | 25 | -93 |
| France | -501 | 803 | -2415 | 1421 | -79 | -1412 | 1601 | -1974 | 5246 | -485 |
| Germany | -420 | 244 | -417 | 310 | 104 | -661 | 628 | -288 | 919 | 179 |
| **Iran** | - | 40 | - | 2 | 9 | - | 79 | - | -93 | 9 |
| Italy | -1082 | 451 | 1804 | 608 | 294 | -2591 | 732 | 1600 | 1999 | 901 |
| **Japan** | - | - | - | 38 | -3 | - | - | - | 92 | -74 |
| **Mexico** | 512 | -82 | - | -99 | -56 | - | -518 | - | -435 | -315 |
| Netherlands | 512 | 172 | 265 | 256 | -21 | - | 363 | 228 | 850 | -47 |
| **Norway** | 29 | 20 | -12 | 12 | 2 | - | 38 | -7 | 45 | 12 |
| **Philippines** | - | -21 | - | | -14 | - | -70 | - | | -30 |
| **Poland** | 20 | -8 | - | | 14 | - | 3 | - | | 2 |
| Portugal | -28 | 41 | 28 | | 4 | -95 | 107 | 24 | | -3 |
| **Romania** | -96 | -1 | - | | -32 | -236 | -2 | - | | -64 |
| **South Africa** | - | - | - | -4 | -8 | - | - | - | -36 | -34 |
| **South Korea** | - | - | - | 7 | 3 | - | - | - | 24 | 11 |
| Spain | 1104 | 167 | -499 | 568 | 317 | 102 | 76 | 712 | 575 | 50 |
| Sweden | 311 | 107 | -102 | -24 | 24 | 693 | 256 | -35 | -301 | -7 |
| Switzerland | 30 | 14 | -306 | | -87 | - | 97 | -275 | | -154 |
| **Turkey** | - | 177 | - | | 118 | - | 394 | - | | 488 |
| United Kingdom | -981 | -3479 | -182 | 226 | -131 | 658 | -3433 | -29 | 1755 | 761 |
| Russia | - | -119 | - | -31 | -155 | - | -199 | - | 229 | -307 |
| India | - | -169 | - | -18 | -75 | - | -630 | - | -241 | -420 |

Table C3: Performance of our model with and without cross-country joint modeling.

| Country features | Correlations | $p$-values |
|---|---|---|
| Cause of death, by communicable diseases and maternal, prenatal and nutrition conditions (% of total) | 0.732615 | 1.25E-06 |
| Mortality rate attributed to unsafe water, unsafe sanitation and lack of hygiene (per 100,000 population) | 0.667864 | 2.17E-05 |
| Incidence of tuberculosis (per 100,000 people) | 0.665138 | 2.41E-05 |
| Prevalence of undernourishment (% of population) | 0.659093 | 3.03E-05 |
| Mortality rate, adult, female (per 1,000 female adults) | 0.648255 | 8.03E-05 |
| Mortality rate attributed to household and ambient air pollution, age-standardized (per 100,000 population) | 0.642676 | 5.51E-05 |
| Mortality rate, under-5 (per 1,000 live births) | 0.60216 | 0.000209 |
| Mortality rate, adult, male (per 1,000 male adults) | 0.570695 | 0.000801 |
| Mortality rate attributed to unintentional poisoning (per 100,000 population) | 0.49254 | 0.003593 |
| Mortality from CVD, cancer, diabetes or CRD between exact ages 30 and 70 (%) | 0.459674 | 0.007117 |
| Population, total | 0.401973 | 0.0204 |
| Birth rate, crude (per 1,000 people) | 0.389252 | 0.025156 |
| Mortality caused by road traffic injury (per 100,000 people) | 0.36984 | 0.03414 |
| PM2.5 air pollution, mean annual exposure (micrograms per cubic meter) | 0.347458 | 0.047564 |
| Urban population (% of total population) | -0.35538 | 0.042399 |
| Current health expenditure per capita (current US$) | -0.37345 | 0.032298 |
| Nurses and midwives (per 1,000 people) | -0.40256 | 0.020202 |
| Current health expenditure (% of GDP) | -0.40361 | 0.019846 |
| GNI per capita, Atlas method (current US$) | -0.40771 | 0.018513 |
| GDP per capita (current US$) | -0.41523 | 0.016262 |
| Immunization, HepB3 (% of one-year-old children) | -0.42523 | 0.024084 |
| Physicians (per 1,000 people) | -0.48562 | 0.005616 |
| Immunization, DPT (% of children ages 12-23 months) | -0.49294 | 0.003562 |
| Immunization, measles (% of children ages 12-23 months) | -0.51081 | 0.002385 |
| Life expectancy at birth, total (years) | -0.57085 | 0.000522 |
| UHC service coverage index | -0.58407 | 0.000359 |
| Prevalence of overweight (% of adults) | -0.61696 | 0.000131 |
| Access to electricity (% of population) | -0.62433 | 0.000103 |
| Cause of death, by non-communicable diseases (% of total) | -0.68116 | 1.28E-05 |
| People with basic handwashing facilities including soap and water (% of population) | -0.8102 | 0.014751 |

Table C4: Correlation between country features and effect of lockdown on $R_0$.

| Country feature | Correlation | $p$-value |
|---|---|---|
| People with basic handwashing facilities including soap and water (% of population) | 0.73285 | 0.038626 |
| Prevalence of overweight (% of adults) | 0.576361 | 0.000447 |
| UHC service coverage index | 0.56557 | 0.000604 |
| Cause of death, by non-communicable diseases (% of total) | 0.546701 | 0.000995 |
| Access to electricity (% of population) | 0.503035 | 0.002847 |
| Life expectancy at birth, total (years) | 0.49019 | 0.003781 |
| Physicians (per 1,000 people) | 0.481126 | 0.006143 |
| Current health expenditure (% of GDP) | 0.448501 | 0.00885 |
| Urban population (% of total population) | 0.429586 | 0.012597 |
| Immunization, measles (% of children ages 12-23 months) | 0.419842 | 0.014999 |

Table C5: Correlation between country features and $R_0$ before lockdown.

| Country | Percentage Difference | Difference |
|---|---|---|
| US | 0.88% | 456 |
| France | 0.14% | 31 |
| United Kingdom | 0.00% | 0 |
| Pakistan | 0.00% | 0 |
| Japan | 0.00% | 0 |
| Italy | 0.00% | 0 |
| Germany | 0.00% | 0 |
| Spain | 0.00% | 0 |
| Belgium | 0.00% | 0 |
| Korea, South | 0.00% | 0 |
| Brazil | 0.00% | 0 |
| Iran | 0.00% | 0 |
| Netherlands | 0.00% | 0 |
| Turkey | 0.00% | 0 |
| Romania | 0.00% | 0 |
| Portugal | 0.00% | 0 |
| Sweden | 0.00% | 0 |
| Switzerland | 0.00% | 0 |
| Ireland | 0.00% | 0 |
| Hungary | 0.00% | 0 |
| Denmark | 0.00% | 0 |
| Austria | 0.00% | 0 |
| India | 0.00% | 0 |
| Ecuador | 0.00% | 0 |
| Russia | 0.00% | 0 |
| Peru | 0.00% | 0 |
| Indonesia | 0.00% | 0 |
| Poland | 0.00% | 0 |
| Philippines | 0.00% | 0 |
| Canada | -0.76% | -18 |

Table C8: The difference in total deaths by April 25 reported at April 25 and May 8. The difference is exactly zero for most countries; for countries with a difference, the difference is negligible percentage-wise. The models in our experiments were trained based on data captured at May 8 while the benchmarks were probably trained using data capture at April 25. As the data are routinely updated, we are expecting to see some occasional retrospective changes (e.g. correcting reporting errors). However, the vast majority of countries are *unchanged* between the two reporting dates, which suggests that our training scheme is valid.

Figure C6: Policy stringency and $R_0$ over time within different countries.

**France**

COVID-19 fatalities

800 deaths
- - - - - - - - - - - - - - - - - - - - - - - - - - - - -

June          July          August

Figure C7: Counterfactual scenario analysis for France.