[Reviews · NeurIPS 2020]

Review 1

Summary and Contributions: The authors present a two-layer GP prior approach to model both fatalities and the effect of various policies on the time-varying reproduction number Rt. The proposed model jointly learns parameters across various countries while still learning random effects of interventions across those countries. The presented model is a semi-mechanistic model that allows to model epidemic based on some epidemiological parameters while still letting the data drive the inference algorithm. They provide a counterfactual analysis for both what was happened and what can happen for assessing the effects of interventions and releasing them subsequently.

Strengths: The authors present a novel two-layer GP model for modeling COVID-19 fatalities. The model in its lower layer combines the compartmental models describing the spread of an epidemic with a GP prior to leverage the data-driven capabilities of GPs to infer parameters for the compartmental models, which in prior literature doesn't exist. The upper layer of the model is another GP that models Rt as a function of various interventions in a country. Although the idea of using interventions to parametrize Rt is not new but using a GP to do it is new and opens up new avenues of learning various counterfactuals based on a different policy enacted in different countries.

Weaknesses: The metric used for evaluation is flawed and gives an unreal sense of better performance. Authors have used cumulative deaths to evaluate their performance which biases the results by taking into account the training fits and numbers which high magnitude. For example, say we have a time series as 10,10,80,20 and we train on first three days. Suppose we have two models with model 1 having its predictions as 10,10,30,30 and the second model as 10,10,70,45. the cumulative error on day 4 is -40 and 15 for models 1 and 2 respectively. However, model 1 has better predicted the held-out or testing point whereas model 2 has calibrated itself to an outlier, hence we can see that using cumulative data to evaluate performance gives a wrong sense of performance. Hence evaluation should be performed on daily deaths. Unfortunately, that will lead the proposed model to have worse performance than the compared models. For example as given in table 2 the performace of Imperial model seems to be very bad for France however that is because of the above stated flaw with evaluation rather than the worse fits. As you can see here https://mrc-ide.github.io/covid19estimates/#/details/France the fits for Imperial model for period april 25-May1/7 are quite good but the loss of performance is on days with some high number of deaths in training data () which were essentially caused due to reporting issues rather than trends. Secondly, given the proposed model is Bayesian and so are the other compared models a more nuanced bayesian performance evaluation is necessary for terms of credible interval width or CRPS, but again on daily deaths, not cumulative deaths. Also, it seems the Imperial model has better performance in Table 1 for short-term prediction but its estimates are not shown for the longer-term prediction. However, from the description of the Imperial model, it seems they can do 14-day prediction too so those estimates should be presented. Another lacking bit is the figures or descriptions of fits for at least a few individual countries for reader to evaluate the goodness of daily fits, again this should be done on daily data not cumulative data for the reasons presented above.

Correctness: The model and infernce scheme is correct however the evaluation is not correct which gives a un real sense of superior performance.

Clarity: Paper is well written and appropriately structured.

Relation to Prior Work: recent Prior work has been adequately discussed but it would help if authors can discuss some olde prior worl especially around the renewal equation literature which undrpins a lot of the compared models.

Reproducibility: Yes

Additional Feedback: Please look at the weakness section and redo the evaluations on daily data. For more reference see https://www.ncbi.nlm.nih.gov/pmc/articles/PMC4426634/ A reason for the low score is how evaluation is not done properly and hence it becomes very difficult to really evaluate the performance of the proposed model. The idea behind the model is still interesting. I have updated my review by firstly increasing the score to 7. However, something I would urge authors to do would be discussing the use of extra features than competing models. I am not stating it is bad but a discussion around it is necessary because a lot of these features were not available when those competing models were released, so it should be acknowledged as performance gain might be due to the use of extra features too.


Review 2

Summary and Contributions: This paper presents a Bayesian model for predicting effects of COVID-19 (in terms of death) across the world in both a global context and localised to understand country and policy specific differences. The algorithm itself encapsulates the policy effects of different countries as input and exploits this to learn a model that can illustrate the effects of different lockdown strategies and their effects on deaths over time [meaning the policies are also allowed to change].

Strengths: The paper's strength is its attempt at balancing a universal and context-specific view of modelling the COVID-19 epidemic. The data-driven nature means the effects of changing policies can be learned and fit by the model.

Weaknesses: It would have been great to also understand the consequence of such a model. What can we do now, outside the evaluation metric of evaluation death prediction in the future. Could we have looked at countries outside US and Italy to look at ways more liberal vs conservative lockdown policies could affect lets say developing countries or countries in the Global South?

Correctness: Methodology as far as this reviewer understand was adequate.

Clarity: The paper is well written. Few typos. You have a typo in table 1. You meant CGP, not CPG

Relation to Prior Work: Work is clearly placed agains prior literature.

Reproducibility: Yes

Additional Feedback: Thank you for taking the time to respond to all the reviewers. Thank you for also highlighting the changes/and or places where examples of your adjustments have been put.


Review 3

Summary and Contributions: This manuscript develops a hybrid Gaussian Process and compartmental epidemiological model for forecasting deaths due to COVID-19 and scenario projections to examine implications of policy decisions.

Strengths: The model formulation is novel. Previous work in the literature has employed a Gaussian Process to allow for time-varying parameters in compartmental models for infectious disease (though I am unfortunately unable to locate a reference at the moment). Previous work has also specified Gaussian Process models where the mean function is a differential equation (e.g. https://arxiv.org/abs/2003.12890) and similar ideas using splines in place of Gaussian Processes (Xun et al. 2013 Parameter Estimation of Partial Differential Equation Models). Finally there exists previous work on models for infectious disease where a flexible non-parametric model for discrepancy is placed on a compartmental model (e.g. Osthus 2019, Dynamic Bayesian influenza forecasting in the United States with hierarchical discrepancy). However, I am not aware of previous work that unifies these ideas.

Weaknesses: The points below from my original review have all been addressed in the author response. Why is forecast accuracy only evaluated at a maximum horizon of 2 weeks? How would the model do at longer horizons? Why weren't evaluations at horizons at least up to 4 included? These horizons are available in the cdc forecast records for the baseline models. I think there are possible justifications for this decision, but it's worth stating them. This is especially relevant since Section 4.3 discusses counterfactual projections made for much larger time horizons, but no attempt has been made to validate the quality of such long-term projections. It would be helpful to give a few more details about the ablation study discussed on lines 243 - 247. Was the exact model described for the hierarchical setting with N=170 countries, including 9 indicators of policies implemented, fit to only the data for the US? If so, does the result really mean that the hierarchical structure is helpful, or maybe just that the full model overfits when fit to data for a single country? If not, what specific changes were made to the model to reduce potential for overfitting when it was applied to a reduced data set? I do believe hierarchical structure should help, it's just not clear what can be said from this study; the leading and concluding sentences of this paragraph seem too strong. In the context of forecasts that may be used to inform public policy decisions, acknowledging and accurately quantifying uncertainty is critical. I would like to see an evaluation of calibration of the model's forecasts. A precise statement of the model is not given. Some specific questions: Lines 122-123: Presumably the observation model for the data {Y_i(t)\ is normal, but this is not directly stated. On lines 140-141, should there also be a variance or amplitude parameter for the kernel?

Correctness: I continue to believe that it is essential to base all forecast evaluation on forecasts generated from the data that would have been available in real time. This is especially important when claiming superiority to baseline models that generated forecasts in real time. The authors indicated in their response that they would do this in the final manuscript. This relates to the following comment from my original review. Line 233 - 234 states that "For each forecast, only data preceding the forecast date was used for training our model." The JHU data are routinely revised, and from the description in Appendix C it sounds like you fit the models to a subset of the data that were available as of May 8. However, in light of revisions to the data, pulling the data that were available at a later date and then subsetting to the data preceding the forecast date is not good enough -- for the comparison with pre-registered forecasts to be reasonable, you need to use the actual data that were available as of the forecast date. If this was done, please clarify that; if not, please correct the analysis. The points below from my original review have been addressed in the author response. Line 233 implies that the error was calculated for forecasts on a daily basis but I did not find forecasts from some of these models at a daily basis; for example, I only found weekly forecasts from the ensemble model in the file at https://www.cdc.gov/coronavirus/2019-ncov/covid-data/files/2020-04-13-model-data.csv. Please clarify how the errors were calculated. Based on that forecast file and the JHU data as of May 8, I was unable to replicate the results for the baseline models in Table 1. In line 233, if in fact you did sum errors for multiple forecasts, it would be good to take the absolute value or square of the error of each forecast before aggregating.

Clarity: In general, the paper was clearly written other than the specific points mentioned above.

Relation to Prior Work: The relation to prior work on forecasting for COVID-19 is clear. There are some gaps in the discussion of literature on Gaussian Processes and compartmental/differential equations models, and Gaussian Processes for infectious disease. I gave some references above. A few additional references that could be added are: - Flaxman et al., Fast hierarchical Gaussian Processes - Johnson et al., PHENOMENOLOGICAL FORECASTING OF DISEASE INCIDENCE USING HETEROSKEDASTIC GAUSSIAN PROCESSES: A DENGUE CASE STUDY (apologies for caps)

Reproducibility: No

Additional Feedback:


Review 4

Summary and Contributions: This submission presents and evaluates a Gaussian process model for characterizing country-specific COVID-19 fatalities as a function of country parameters and lockdown policies. The paper presents a novel model and evaluates it by comparing its forecasts against those of several widely used models. The paper represents an important and technically sound application of Gaussian process methodology to better understanding and controlling the spread of COVID-19.

Strengths: The strengths of the submission and the presented approach include the following: - The approach is able to answer counterfactual queries based on varied lockdown policies. - The prior mean function is based on the well established SEIR model. - The approach is able to characterize countries with relatively little data by representing R_0 as a function of country-specific variables. - The approach is empirically compared to the range of commonly used models for COVID-19 forecasting. - The presented experiments demonstrate the value of the model in forecasting COVID-19 deaths. - The submission is very clearly written.

Weaknesses: Given space, the paper could be strengthened by discussing insights that can be garnered by inspecting the learned model. E.g. which country variables seem to be the most/least important for determining R_0?

Correctness: Yes, both the methodology and empirical evaluation are technically sound.

Clarity: Yes, the writing is excellent. One typo is "each new patient infect" -> "each new patient infects"

Relation to Prior Work: The discussion or prior work is sufficiently clear and comprehensive.

Reproducibility: Yes

Additional Feedback:

[Author Response · NeurIPS 2020]

[[ **Reviewer 1** ]] Thank you for the excellent comments and suggestions; we have updated the paper after taking all your comments into account. The 2-week performance of Imperial model for the US was mistakenly missed in Table 1, it is now provided in Updated Table 2. ■ **Evaluation metric:** We agree that evaluation on daily deaths is a more accurate metric for a model's generalization performance. We have amended Tables 1 and 2 by replacing the accuracy of predicting cumulative deaths with that of daily (incident) deaths—the updated results are summarized in Updated Table 2. With the new metric, our model still outperforms the baselines in the same countries and performs competitively in countries where it is not the best. More importantly, our key conclusions and insights regarding global hierarchical modeling are still preserved under the new metric. ■ **Uncertainty intervals:** Based on your suggestion, we evaluated the average *continuous ranked probability score* (CRPS) on daily deaths in Updated Table 2. Our model's probabilistic forecasts performed competitively compared to the baselines in all countries; we will also add results on coverage probabilities and CI length in the final version of the paper. ■ **Figures:** Fig. 3 (b) depicted the goodness of fit for daily deaths in the UK. In the final version of the paper, we will use the extra space to add similar figures for all countries in Table 2.

**Updated Table 2**: Accuracy of daily deaths predicted by baselines. (The Imperial model does not provide 30-day forecasts.)

| Country | Mean Absolute Error on Daily Deaths (CRPS: continuous ranked probability score) | | | | | | |
|---|---|---|---|---|---|---|---|
| | 14-day Forecasts | | | | 30-day Forecasts | | |
| | CGP | Imperial | IHME | YYG | CGP | IHME | YYG |
| US | 139 (0.076) | 149 (0.282) | 753 (0.164) | **50** (**0.073**) | 481 (0.196) | 957 (0.260) | **365** (**0.164**) |
| UK | **58** (0.089) | 164 (0.248) | 288 (**0.088**) | 178 (0.224) | 231 (0.291) | 259 (**0.156**) | **140** (0.176) |
| Italy | 78 (**0.090**) | **63** (0.226) | 202 (0.298) | 87 (0.192) | **55** (**0.119**) | 179 (0.324) | 90 (0.184) |
| Germany | **30** (**0.100**) | 51 (0.247) | 54 (0.151) | 70 (0.249) | **45** (**0.197**) | 46 (0.230) | 91 (0.273) |
| Spain | 125 (**0.121**) | 88 (0.236) | 133 (0.197) | **82** (0.183) | 83 (**0.168**) | 140 (0.273) | **81** (0.170) |
| France | **26** (**0.075**) | 85 (0.239) | 148 (0.216) | 124 (0.161) | **104** (0.190) | 150 (0.282) | 153 (**0.170**) |
| Netherlands | **11** (0.131) | 29 (0.298) | 83 (**0.112**) | 34 (0.220) | **32** (0.277) | — | 45 (**0.241**) |
| Sweden | **11** (0.098) | 34 (0.271) | 35 (**0.082**) | 32 (0.218) | **34** (**0.210**) | 118 (**0.210**) | 38 (0.228) |
| Portugal | **1** (**0.092**) | 2 (0.176) | 7 (0.186) | 10 (0.260) | **3** (**0.174**) | 10 (0.275) | 12 (0.263) |

[[ **Reviewer 2** ]] Thank you for your feedback. We will fix the typo in Table 1. ■ **Broader consequences:** We agree that the model can be used to analyze liberal/conservative lockdown policies in developing countries. In fact, Table C4 and Table C5 in the Appendix already present an analysis on how country features impact the effectiveness of lockdown. We have collected more data since the time of submission and will update and augment this analysis in the final manuscript. Moreover, the model can be used to conduct counterfactual analysis as shown in Fig. 3.

[[ **Reviewer 4** ]] Thank you for the excellent comments and valuable suggestions. We will include all the suggested references in the final version of the paper. We would like to clarify that our model was trained on the archived data capture of May 8; in the final manuscript, we will also add a robustness analysis to examine the model performance on subsequent data updates. ■ **Long-term forecasts:** We focused on 2-week forecasts to enable comparisons with all baselines as some of the benchmarks do not issue long-term predictions (e.g., the Imperial model). As shown in Updated Table 2, our model performs equally well when tested on 30-day forecasts; it provides the same patterns of accuracy gains achieved on the 2-week forecast. ■ **Evaluating uncertainty measures:** We evaluated the quality of our probabilistic forecasts in terms of the average continuous ranked probability score (CRPS) in Updated Table 2. Please also refer to Lines 8-11 of our response to Reviewer 1. ■ **Evaluation metric:** We apologize for the typo in Line 233—in the original submission, accuracy was evaluated on predicted *cumulative* deaths rather than *incident* deaths. This is why we were able to evaluate the accuracy of the weekly forecast by the CDC-ensemble. In Updated Table 2, we evaluate the performance of all baselines with respect to the mean *absolute* error in the predicted daily deaths, i.e., $\mathcal{E} = \frac{1}{T} \sum_{k=1}^{T} |Y_i(t+k) - \widehat{Y}_i(t+k)|$. We will release the code for reproducing Updated Table 2. ■ **Model specification:** We use a standard radial basis function (RBF) kernel with a variance (amplitude) parameter. The data $Y_i(t)$ is assumed to be normal. We will provide the precise expression of the the distribution of $Y_i(t)$ and expand the kernel parameter set in lines 122 and 140 of the revised manuscript. ■ **Ablation study:** Your description of our ablated baseline is accurate; we will clarify the details in the final paper. The benefits of hierarchical modeling are multifaceted: (a) policy heterogeneity across countries regularizes *factual* fits enabling better generalization on *counterfactual* inferences, (b) asynchronicity of the pandemic across countries enables better generalization *over time* for lagging countries, and (c) countries with similar features share the epidemic parameters. While it is hard to disentangle these effects analytically, we will add more ablated baselines with clusters of countries (with similar policies to the US, similar features to the US, and pandemic onsets synchronized with the US) removed one at a time to empirical assess these effects separately.

[[ **Reviewer 5** ]] Thank you for your feedback. We will fix the typo in Line 61. ■ **Model Inspection:** Table C4 and Table C5 in the Appendix already show the ranking of country features with respect to their impact on $R_0$. Based on your suggestion, we will move these results to the main manuscript given the extra space allowed in the final manuscript.

[Meta-Review · NeurIPS 2020]

Four knowledgable reviewers feel the paper is of high technical quality, novel, and the claims are well supported by empirical evidence; all of this puts it comfortably above the bar for publication at NeurIPS. R1 and R4 raised some technical concerns, primarily about evaluation in their initial reviews (evaluate on incident instead of cumulative deaths, use longer time horizons, and evaluate probabilistic calibration). The authors included these results in their rebuttal; the reviewers were convinced by this additional evidence and raised their scores. The authors are encouraged to take reviewer comments into account in the final version, including: acknowledge the use of additional features compared to the baseline models (R1), include additional related work (R3), clarify the scheme used to snapshot data for hindcasting and justify the fairness of the comparison to pre-registered forecasts (R3).